# Effect of Greenhouse CO$_2$ Supplementation on Yield and Mineral Element Concentrations of Leafy Greens Grown Using Nutrient Film Technique

**Hardeep Singh [1]** , **Megha R. Poudel [1,\*]**, **Bruce L. Dunn [1]**, **Charles Fontanier [1] and Gopal Kakani [2]**

[1] Department of Horticulture and Landscape Architecture, Oklahoma State University, Stillwater, OK 74078-6027, USA; hardeep.singh@okstate.edu (H.S.); bruce.dunn@okstate.edu (B.L.D.); charles.fontanier@okstate.edu (C.F.)

[2] Department of Plant and Soil Sciences, Oklahoma State University, Stillwater, OK 74078, USA; v.g.kakani@okstate.edu

\* Correspondence: mpoudel@ufl.edu

**Abstract:** Carbon dioxide (CO$_2$) concentration is reported to be the most important climate variable in greenhouse production with its effect on plant photosynthetic assimilation. A greenhouse study was conducted using a nutrient film technique (NFT) system to quantify the effect of two different levels of CO$_2$ (supplemented at an average of 800 ppm and ambient at ~410 ppm) on growth and nutritional quality of basil (*Ocimum basilicum* L.) 'Cardinal', lettuce (*Lactuca sativa* L.) 'Auvona', and Swiss chard (*Beta vulgaris* L.) 'Magenta Sunset' cultivars. Two identical greenhouses were used: one with CO$_2$ supplementation and the other serving as the control with an ambient CO$_2$ concentration. The results indicate that supplemented CO$_2$ could significantly increase the height and width of hydroponically grown leafy greens. Supplemented CO$_2$ increased the fresh weight of basil 'Cardinal', lettuce 'Auvona', and Swiss chard 'Magenta Sunset' by 29%, 24.7%, and 39.5%, respectively, and dry weight by 34.4%, 21.4%, and 40.1%, respectively. These results correspond to a significant reduction in Soil Plant Analysis Development (SPAD) and atLEAF values, which represent a decrease in leaf chlorophyll content under supplemented CO$_2$ conditions. Chlorophyll, nitrogen (N), phosphorus (P), and magnesium (Mg) concentrations were generally lower in plants grown in supplemented CO$_2$ conditions, but the results were not consistent for each species. Supplemented CO$_2$ reduced tissue N concentration for basil 'Cardinal' and lettuce 'Auvona' but not Swiss chard, while Mg concentration was reduced in supplemented CO$_2$ for Swiss chard 'Magenta Sunset' only. In contrast, Fe concentration was increased under supplemented CO$_2$ for basil 'Cardinal' only. These findings suggest CO$_2$ supplementation could increase yield of leafy greens grown with hydroponics and have varying impact on different mineral concentrations among species.

**Keywords:** soilless culture; lettuce; basil; Swiss chard; SPAD; atLEAF; mineral concentration

## 1. Introduction

With an increasing human population, the demand for food is also increasing, while the arable land per capita is being reduced throughout the world [1]. Protected soilless culture can be a good approach to help in sustainable production for feeding the increasing population. Soilless culture can be defined as cultivation of nonaquatic plants without use of mineral soil as a growth substrate, while all essential plant nutrients are provided through a nutrient solution [2]. In recent years, different systems of soilless culture (e.g., hydroponics, aeroponics, gravel culture, and rockwool culture) have been adopted worldwide for food production [3]. Cleaner and longer postharvest life of hydroponic

produce could be one of the reasons for this increase [4,5]. Soilless culture has also been viewed favorably for its greater efficiency of water use, due to lack of loss from runoff, infiltration, evaporation from soil [6], and increased nutrient efficiency from recycling the nutrient solution [7]. The nutrient film technique (NFT) developed by Allen Cooper and his colleagues in 1960 is among the more popular hydroponic systems for cultivation of leafy greens [8].

A soilless culture production system can be profitable, particularly if the grower maintains year-round production [9]. However, year-round production of green leafy vegetables may require supplemental lighting in the winter and shade in the summer [4]. Carbon dioxide supplementation can be equally beneficial as light supplementation due to its effect on plant photosynthetic assimilation [10]. However, $CO_2$ supplementation is most beneficial in autumn, spring, and winter when the ventilation system is closed [11]. Mortensen [10] reported that the advantages of $CO_2$ enrichment in the greenhouse atmosphere were detected as early as the nineteenth century, but the technique was not used commercially until the 1960s when both cheap sources of high-purity $CO_2$ and gas-tight greenhouse constructions became available [10,12].

A study showed that the $CO_2$ concentration inside a sealed greenhouse can be as low as 150 ppm during the day, and $CO_2$ supplementation was required in winter for optimum production of cucumber (*Cucumis sativus* L.) [13]. Similarly, a 30% increase in photosynthesis assimilation and growth of lettuce (*Lactuca sativa* L.) was reported as a result of $CO_2$ supplementation in the greenhouse environment [9,10]. Furthermore, Both and colleagues [9] reported that a $CO_2$ level of 400–600 μmol mol$^{-1}$ is suitable for hydroponic culture of lettuce. In addition to the benefits of increased production, $CO_2$ supplementation is reported to reduce transpiration in lettuce [4].

Greenness of leafy vegetables is an indicator of carotene content [14] as well as an influential trait affecting consumer preference and acceptability [15]. It was reported that $CO_2$ supplementation can result in increased or decreased leaf chlorophyll content of leaves [16,17]. At the same time, meta-analysis conducted by Dong et al. [18] reported that $CO_2$ supplementation had no effect on chlorophyll concentration. Similarly, when basil (*Ocimum basilicum* L.) was grown under 1500 ppm $CO_2$, interveinal chlorosis was observed due to the accumulation of large grains of starch [19]. Chlorophyll meter readings using a Soil Plant Analysis Development (SPAD) meter can serve as an indicator of greenness for green leafy vegetables [20]. In addition, these chlorophyll meters can serve as good non-destructive indicators of nutrient content of leafy greens as significant correlations between chlorophyll meter readings and nutrients like nitrogen (N), potassium (K), and phosphorus (P) have been reported in the literature [21–23]. Another important characteristic of leafy green vegetables is their nutritional quality, which is also reported to be affected by supplementation of $CO_2$ in the growth environment. It is reported that $CO_2$ supplementation results in a decrease in nitrate concentrations of about 26% and 19% in fruit and leafy vegetables, respectively [16].

Humans have been consuming herbs for thousands of years. In the past as well as today, herbs are used for cosmetic, medicinal, and culinary purposes [24]. Tyson et al. [25] reported that herbs cover more than 18% of greenhouse acreage in the United States and were the third most grown crop in greenhouses. Similarly, year-round productions of leafy greens in greenhouses are gaining in popularity in the United States [26]. Although many studies have covered herbs and leafy greens in greenhouse production, very few have explored how greenhouse $CO_2$ supplementation affects the growth and nutritional quality of these crops. Therefore, the objective of this study is to determine the effect of greenhouse $CO_2$ supplementation on yield and nutritional quality of lettuce, basil, and Swiss chard (*Beta vulgaris* L.) grown hydroponically in an NFT system.

## 2. Materials and Methods

### 2.1. Plant Material and Growth Conditions

Seeds of 'Auvona' open-heart romaine lettuce, 'Magenta Sunset' Swiss chard, and 'Cardinal' basil were obtained from Johnny's selected seeds (Winslow, ME, USA). Seeds were sown in rockwool starter

cubes (1.5 cm$^3$) with 98 cubes to a sheet (Grodan, Milton, Ontario, Canada) on 20 January 2016 and were transplanted into NFT tables (Hydrocycle 4-inch Pro NFT series system; Growers Supply, Dyersville, IA, USA) on 25 February 2016 (35 days after seeds were sown) at the Oklahoma State University (OSU) Department of Horticulture and Landscape Architecture Research Greenhouses (Stillwater, OK, USA). Each table had 10 channels measuring 10 cm wide, 5 cm deep, and 366 cm long. Channel lids had 18 predrilled circular holes 3.5 cm in size spaced 20.3 cm on center. One transplant was placed in each slot with 15 plants per species per table. The NFT channels had a slope of 2.8% between the irrigation and drainage end, and the water flowing along this slope was collected in a tank and recirculated by a pump to the irrigation pipe. The experiment was repeated with an additional planting of seedlings on 28 February 2017.

Hydroponic fertilizer 5N-4.8P-21.6K (Peter's, J.R. Peters Inc., Allentown, PA, USA) and calcium nitrate (Haifa North America, Altamonte Spring, FL, USA) were used as fertilizer [24]. Tap water with an electrical conductivity (EC) of 0.5 dS m$^{-1}$ and a pH of 7.8 was used to prepare the nutrient solution. In the tank, 147.41 g of 5N-4.8P-21.6K and 97.5 g of calcium nitrate were added to make 150 ppm of N. In 2-week intervals, the tanks were flushed and refilled to remove the excess nutrient buildup. The EC of all the nutrient solutions was maintained at 1.5–2.5 dS m$^{-1}$, and the pH was maintained at 5.5 to 6.5 as recommended by Singh and Dunn [27]. The pH and EC of each solution was checked and maintained every third day. The nutrient solution pH was maintained using pH up and pH down solutions (General Hydroponics, Santa Rosa, CA, USA), whereas EC was maintained by adding water if the EC was high and adding nutrient solution in the same proportion of both bags if the EC was less than the recommended limit.

## 2.2. Experimental Setup

The study was conducted in a split-plot design. Two identical greenhouses were used and one of the greenhouses was fitted with a natural gas-burning $CO_2$ generator (Johnson Gas Appliances, Cedar Rapids, IA, USA) in the middle of the greenhouse. The $CO_2$ generator was set to produce a daily average of 800 ppm of $CO_2$ by burning natural gas (Figure 1) during supplementation period. The generator was automatic and turned on from 6:00 a.m. to 14:00 p.m. A $CO_2$ monitor (FLIR Commercial System Inc., Nashua, NH, USA) monitored the $CO_2$ concentration in both greenhouses. Both greenhouses were set at 21/18 °C day/night temperature and exposed to natural photoperiod resulting in a daily light integral of 12 to 14 mol m$^{-2}$ d$^{-1}$ as measured using a data logger (T & D Corporation, Nagano, Japan). The average relative humidity for the greenhouse was 28%. Each species with $CO_2$ treatment had 15 replicate plants. Similar methods described above were followed for the second study.

## 2.3. Data Collection

Data were collected 46 days after transplanting of plugs in the NFT system. Each plant was scanned using two different chlorophyll meters (SPAD-502, Spectrum Technologies, Aurora, IL, USA; and atLEAF, FT Green, Wilmington, DE, USA) at the time of harvest. For each plant, SPAD and atLEAF readings were taken from three mature leaves representing the base, middle, and top of the plant. For each plant, the SPAD and atLEAF readings were recorded as the average of single readings at the tip, base, and blade leaves of a plant. Plant height (from top of the table to plant tip), diameter (average of diagonal width), specific leaf area (SLA), total leaf area, fresh weight, dry weight, and plant mineral element concentrations were measured. Total leaf area was measured using a LI-3000C area meter (LI-COR, Inc., Lincoln, NE, USA). Specific leaf area was calculated as the ratio of one-sided total leaf area to the total dry weight of a plant. For each species in an experimental unit, three samples were taken for leaf area measurement and the same samples were used for mineral element concentrations analysis. After area measurements, leaves were dried in an oven at 57 °C for 72 h to measure dry weight. The samples were then sent to the Soil, Water and Forage Analytical Laboratory at Oklahoma State University for analysis of leaf mineral element concentrations using a nutrient

analyzer (TruSpec Carbon and Nitrogen Analyzer; LECO Corp., St. Joseph, MI, USA). For mineral element concentration, the plant material was digested on a digestion block at 115 °C with concentrated nitric acid. The resulting solution was analyzed for mineral element concentrations on an inductively coupled plasma spectrometer.

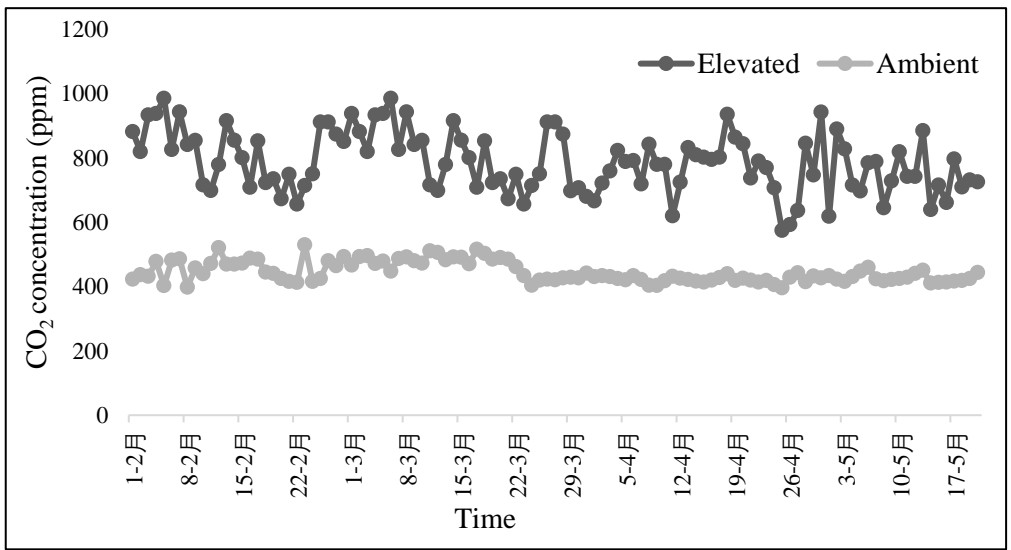

**Figure 1.** Average daily value of ambient and supplemental $CO_2$ concentration measured at Oklahoma State University (OSU) Department of Horticulture and Landscape Architecture Research Greenhouses (Stillwater, OK, USA) during the study period (average of 2016 and 2017).

*2.4. Statistical Analysis*

The experiment was analyzed as a split-plot design repeated in time. The whole main plots were two $CO_2$ concentrations (~400 and an average of 800 ppm) and subplots were assigned from three species (lettuce, basil, and Swiss chard). Statistical analysis was performed at $p > 0.5$ using SAS/STAT software (version 9.4; SAS Institute, Cary, NC, USA). Data were subjected to PROC MIXED and pdmix800, a macro program used to compute means. To compare differences between treatment means the Tukey–Kramer test was used.

**3. Results**

*3.1. Basil*

Under supplemented $CO_2$ conditions, both the height and width of 'Cardinal' were greater as compared to ambient $CO_2$ conditions (Table 1). Similarly, fresh weight of 'Cardinal' was also greater in supplemented $CO_2$ conditions by 24.7%, over ambient $CO_2$ conditions (Table 1). As a result, the dry weight of 'Cardinal' was also greater under supplemented $CO_2$ conditions. 'Cardinal', grown under supplemented $CO_2$ conditions, was greater in size based on leaf number (data not shown). Carbon dioxide supplementation resulted in a significant increase of total leaf area of 'Cardinal' by 41.9% (Table 1). The SLA for 'Cardinal' was greater in ambient $CO_2$ conditions (260.7 $cm^2$ $g^{-1}$) as compared to supplemented $CO_2$ conditions (160 $cm^2$ $g^{-1}$) (Table 1). Similarly, SPAD and atLEAF values in ambient $CO_2$ conditions were greater by 5.1% and 6.2% over supplemented $CO_2$ conditions, respectively (Table 1). The N concentration was lower in 'Cardinal' leaves produced under supplemented $CO_2$ conditions, while the Fe concentration was greater (Table 2).



**Table 1.** Effect of $CO_2$ treatments (ambient at 400 ppm and supplemented at an average of 800 ppm) on height, width, fresh weight, dry weight, Soil Plant Analysis Development (SPAD), atLEAF, total leaf area, and specific leaf area values for lettuce 'Auvona', Swiss chard 'Magenta Sunset', and basil 'Cardinal' grown under nutrient film technique (NFT) in Stillwater, OK, USA in 2016 and 2017.

| Carbon Dioxide | Height (cm) | Width (cm) | Fresh Weight (g) | Dry Weight (g) | SPAD (unitless) | atLEAF (Unitless) | Total Leaf Area (cm²) | Specific Leaf Area [z] (cm² g⁻¹) |
|---|---|---|---|---|---|---|---|---|
| | | | | Basil | | | | |
| Ambient | 34.0b [y] | 26.2b [y] | 123.1b [y] | 11.9b [y] | 44.4a [y] | 52.6a [y] | 1644.4b [x] | 260.7a [x] |
| Elevated | 36.9a | 29.8a | 158.9a | 15.0a | 42.2b | 49.6b | 2333.8a | 160.0b |
| | | | | Lettuce | | | | |
| Ambient | 26.6a | 25.2a | 203.8b | 19.1b | 48.5a | 45.8a | 4884.5b | 271.1b |
| Elevated | 26.5a | 25.7a | 254.2a | 23.8b | 45.1b | 48.6a | 5988.5a | 321.6a |
| | | | | Swiss chard | | | | |
| Ambient | 48.4b | 33.3b | 296.8b | 26.0b | 50.6a | 54.8a | 2836.4b | 105.9a |
| Elevated | 52.8a | 35.4a | 414.1a | 38.4a | 47.4b | 51.0b | 3801.1a | 105.8a |

[z] Specific leaf area is the ratio of leaf area of one side of an individual leaf to the dry weight of the same leaf; [y] Means ($n = 30$) within a parameter of an individual species followed by the same letter are not significantly different at $p \leq 0.05$; [x] Means ($n = 10$) within a parameter of an individual species followed by the same letter are not significantly different at $p \leq 0.05$.

**Table 2.** Effect of $CO_2$ treatments (ambient at 400 ppm and supplemented at an average of 800 ppm) on mineral element concentrations of lettuce 'Auvona', Swiss chard 'Magenta Sunset', and basil 'Cardinal' grown under nutrient film technique (NFT) in Stillwater, OK, USA in 2016 and 2017.

| Carbon dioxide | Nitrogen (%) | Phosphorus (%) | Calcium (%) | Potassium (%) | Magnesium (%) | Sulphur (%) | Boron (ppm) | Manganese (ppm) | Iron (ppm) | Zinc (ppm) |
|---|---|---|---|---|---|---|---|---|---|---|
| Basil | | | | | | | | | | |
| Ambient | 5.1a [z] | 0.6a | 2.6a | 2.97a | 0.86a | 0.30a | 36.70a | 57.96a | 135.38b | 49.90a |
| Elevated | 4.6b | 0.5a | 2.5a | 3.21a | 0.84a | 0.29a | 40.46a | 55.93a | 239.30a | 48.75a |
| Lettuce | | | | | | | | | | |
| Ambient | 4.3a | 0.6a | 1.8a | 5.68a | 0.63a | 0.28a | 49.93a | 99.90a | 154.40a | 38.90a |
| Elevated | 3.7b | 0.4b | 1.6a | 5.12a | 0.81a | 0.26a | 47.51a | 95.20a | 195.01a | 32.45a |
| Swiss chard | | | | | | | | | | |
| Ambient | 4.6a | 0.3b | 1.7a | 4.38a | 1.14a | 0.37a | 61.93a | 108.03a | 97.01a | 35.75a |
| Elevated | 4.4a | 0.5a | 1.4a | 4.55a | 0.70b | 0.33a | 59.68a | 71.08a | 137.55a | 39.96a |

[z] For each species, means ($n = 6$) within a column with the same letters are not significantly different at $p \leq 0.05$.

### 3.2. Lettuce

There was no significant difference in height and diameter of 'Auvona' between different $CO_2$ treatments (Table 1). Fresh weight of 'Auvona' was greater (24.7%) in supplemented $CO_2$ conditions as compared to ambient $CO_2$ conditions (Table 1). 'Auvona' plants were compact and weighed more but were of equal size in visual appearance (Figure 2). However, a physiological disorder of tipburn on inner leaves at a later growth stage was observed under supplemented $CO_2$ conditions, while the plants under ambient conditions were healthy. Total leaf area for 'Auvona' was also greater (22.6%) under supplemented $CO_2$ conditions as compared to ambient $CO_2$ conditions (Table 1). Therefore, the SLA of 'Auvona' was greater in supplemented $CO_2$ and was 271.1 and 321.6 cm$^2$ g$^{-1}$ in ambient and supplemented $CO_2$, respectively (Table 1). The SPAD values for 'Auvona' were greater in ambient $CO_2$ conditions as compared to supplemented $CO_2$ conditions (Table 1). However, there was no significant difference in atLEAF values between different $CO_2$ treatments. For foliar mineral element concentrations, N and P concentrations were greater in ambient $CO_2$ conditions, while there was no significant difference among the two $CO_2$ treatments for concentrations of other mineral elements (Table 2).

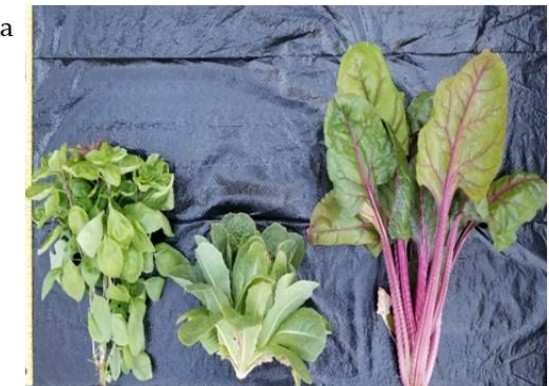 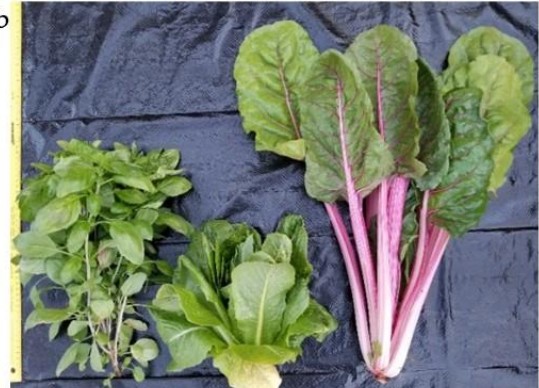

**Figure 2.** Basil 'Cardinal', lettuce 'Auvona', and Swiss chard 'Magenta Sunset' (left to right in both figures) grown in (**a**) ambient and (**b**) supplemented $CO_2$ condition 46 days after transplanting.

### 3.3. Swiss Chard

For 'Magenta Sunset', $CO_2$ supplementation also resulted in increased height and plant width (Table 1). A greater (39.5%) fresh weight under supplemented $CO_2$ conditions was observed between ambient (296.8 g) and supplemented $CO_2$ (414.1 g) conditions (Table 1). Due to the greater number of leaves and greater plant size, the total leaf area of 'Magenta Sunset' also increased by 34% under supplemented $CO_2$ conditions (Table 1). In contrast to lettuce 'Auvona' and basil 'Cardinal', there was no significant difference in the SLA in 'Magenta Sunset'. The SPAD and atLEAF values for 'Magenta Sunset' were greater under ambient $CO_2$ conditions in comparison to supplemented $CO_2$ conditions (Table 1). Among the different foliar mineral element concentrations, P and Mg concentrations were greater under ambient $CO_2$ conditions as compared to supplemented $CO_2$ conditions (Table 2).

## 4. Discussion

During winters, greenhouses are not ventilated in order to keep them warmer; may result in depletion of greenhouse $CO_2$ concentrations below ambient $CO_2$ concentrations and suppression of photosynthesis and growth of vegetables [28,29]. Therefore, if greenhouses are supplemented with $CO_2$ during this period it can result in increased growth rate due to increased photosynthesis [30]. Similar to the present study, the growth rate of lettuce was reported to increase by 30% under supplemented $CO_2$ conditions, presumably due to an increase in the rate of photosynthetic assimilation [31]. However, the response of $C_3$ plants in terms of photosynthetic acclimation is specific [32] and shows a positive

response up to a certain concentration of $CO_2$ only. Above 800–1000 ppm, some species may reach a saturation point and net photosynthesis does not increase with increasing $CO_2$ [33].

Supplemented $CO_2$ increases carbohydrate sink size, which results in increased photosynthetic accumulation and vegetative growth of different crops. The biomass and dry matter production was also expected to increase due to increased photosynthetic assimilation and growth rate in all three species. As expected, all three species under supplemented $CO_2$ showed a significant increase in fresh and dry matter production. Similarly, previous studies also reported an increase in dry weight production under supplemented $CO_2$ conditions in hydroponically grown lettuce [8,34], basil [32], and greenhouse grown Swiss chard 'Fordhook Giant' [35] with $CO_2$ concentrations of 1300 ppm, 1500 ppm, and 72.5 ± 2.2 Pascal, respectively.

Most prior $CO_2$ related studies reported a decrease in the SLA of a plant. Due to the storage of starch in leaves, the leaves of peanuts (*Arachis hypogaea* L.) had greater dry weight at 800 and 1200 ppm as compared to ambient (400 ppm) and resulted in a higher SLA [33]. However, Harmens et al. [36] explained that a decrease in SLA simply cannot be explained through increased photosynthesis and accelerated growth of plants under supplemented $CO_2$. Rather, SLA depends on how assimilates are distributed in shoots and roots during various growth stages. Thus, considering both root and shoot parameters in future studies will help in understanding species specific nature of partitioning of assimilates in the roots and shoots.

Similar to the present study, Gillig et al. [32] also reported a significant decrease in chlorophyll level in hydroponically grown basil when grown at 400 and 1500 ppm $CO_2$. Similarly, development of interveinal chlorosis was observed due to the accumulation of large grains of starch in basil grown under 1500 ppm $CO_2$ concentration [17]. The chlorophyll level under supplemented $CO_2$ can be explained by movement of N to other sinks or it may be due to degradation of the chlorophyll [37]. Another possible reason explaining a decrease in chlorophyll content is accumulation of non-structural carbohydrates under supplemented $CO_2$ [38]. These non-structural carbohydrate accumulations are generally thought to physically distort the chloroplast [39].

Tissue N concentration of above-ground tissue is reported to be lowered by 10–15% as a result of $CO_2$ supplementation in many species [40,41]. Studies related to leaf nutrient content in cotton (*Gossypium hirsutum* L.) [42], chrysanthemum (*Chrysanthemum × morifolium* Ramat.'Fiesta') [43], and hydroponically grown lettuce ('Mantilla') [44] reported a decrease in leaf N and P concentrations, which corresponds to results in our study, but the response was inconsistent among species. A robust single mechanism for lower nutrient content under supplemented $CO_2$ has not yet been developed. The hypothesis has been described in previous studies which include dilution of N due to increased carbohydrates, decreased N uptake, decreased N demand, and reduced transpiration [45] of crops [41,46]. It was reported that $CO_2$ supplementation limits the uptake of N and the synthesis of nitrogenous compounds of vegetables by 9.5% [47]. In the literature, it was reported among different mineral elements that Fe concentration experienced the greatest decrease (31%) in leafy greens grown under supplemented $CO_2$ conditions [17]. In the current study, contrasting results were seen for basil 'Cardinal' where Fe concentration increased significantly in $CO_2$ supplemented conditions. Similarly, an increase in leaf Fe concentration in hydroponically grown lettuce ('Mantilla') was reported under supplemented $CO_2$ conditions [44]. The possible explanation for this increase in Fe concentration in some cases could be an increase in root nitric oxide levels under Fe-limited and elevated $CO_2$ conditions [48]. Nitric oxide was indicated as a signal molecule involved in playing a role in regulating gene expression during Fe deficiency [49]. It is possible that the Fe concentration in the nutrient solution may have gone below the basil requirement and increased nitric oxide in roots may have upregulated Fe acquisition response under supplemented $CO_2$ conditions. However, an increase in Fe concentration of leafy greens when grown in supplemented $CO_2$ has beneficial effects for human nutrition [50]. Duval et al. [46] reported that the effect of supplemented $CO_2$ on plant nutrient content depends on available N, tissue type, species, and nutrient ions. Some nutrients respond well, and some are not affected by supplemented $CO_2$.

Tipburn is a physiological disorder in lettuce which is generally associated with distribution of calcium (Ca) ions within plant leaves [51]. A lower rate of transpiration and high humidity are environmental factors affecting Ca uptake and cause localized tipburn. The use of horizontal air flow (HAF) fans in the greenhouse can be a potential solution to decrease tip burn losses by increasing transpiration and lowering relative humidity. Another reason for the development of necrotic brown spots in the margin of developing leaves is an inability to meet Ca demand of a quickly growing plant (due to supplemented $CO_2$, high light intensity, and greater fertilizer rate) causing lower distribution of Ca to inner leaves [52]. Additionally, Gilliham et al. [53] reported that translocation of Ca in plants is predominated by an apoplastic pathway and rate of transpiration determines the Ca concentration in plant tissue. The distribution of Ca in plant tissue is heterogeneous and the concentration of Ca might differ between inner and outer leaves depending upon the growing environment [54]. Plants grown with supplemental $CO_2$ show lower rates of transpiration due to reduced stomatal conductance. A lower rate of transpiration might have resulted in a lower Ca concentration in inner leaves resulting in tipburn [8]. Although there was no significant difference in Ca concentration of plants grown in ambient and supplemented $CO_2$ during whole plant mineral elements analysis, there might be a difference in inner and outer leaf Ca concentrations, which was not considered during the study. Since environmental factors (lower transpiration and higher humidity) could be the cause of tipburn under supplemented $CO_2$; vertical air flow within greenhouses could be a feasible solution for the tipburn problem [52].

## 5. Conclusions

Results suggest that supplemented $CO_2$ has significant potential to increase growth and development of leafy greens grown in NFT systems. Increased growth rate could result in early harvest and more crop cycles each year and thereby help in feeding the increasing world population. The growth response of different species varied, but this study showed increased growth of all three species. Supplementing $CO_2$ in greenhouse environments during growth of hydroponically grown leafy greens may also result in lighter green (due to low chlorophyll content) produce which may impact the marketability of the produce. Physiological disorders such as tipburn in 'Auvona' may also reduce produce quality when grown under supplemented $CO_2$ conditions. For mineral concentrations, the study suggests that $CO_2$ supplementation may have both a positive and negative effect as lower leaf N concentration might affect available protein, while greater Fe concentration in our food when grown with a nutrient solution containing 2.30 ppm of Fe is a desired quality. Thus, future studies should examine the nutritional aspect and physiological changes in nutrient and water uptake of crops grown in supplemented $CO_2$ conditions and what role HAF fans and air movement have on overall plant quality.

**Author Contributions:** Conceptualization, B.L.D. and M.R.P.; methodology, M.R.P.; software, M.R.P.; validation, H.S.; formal analysis, M.R.P.; investigation, B.L.D. and H.S.; resources, B.L.D.; writing—original draft preparation, M.R.P.; writing—review and editing, B.L.D.; H.S.; C.F.; G.K.; supervision, B.L.D.; project administration, B.L.D.; funding acquisition, B.L.D. All authors have read and agreed to the published version of the manuscript.

**Funding:** This work was supported by the USDA National Institute of Food and Agriculture, Hatch project, and the Division of Agricultural Sciences and Natural Resources at Oklahoma State University.

**Acknowledgments:** The authors are thankful to Stephen Stanphill for his support in conducting experiments at greenhouse.

**Conflicts of Interest:** The authors declare no conflicts of interest.

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
