# Peer review of "Effect of Greenhouse CO2 Supplementation on Yield and Mineral Element Concentrations of Leafy Greens Grown Using Nutrient Film Technique"

_agronomy, doi:10.3390/agronomy10030323_

Round 1
Reviewer 1 Report
Report on Agronomy 717865
A reasonably written manuscript, but with a lot of minor mistakes that must be attended to before possible publication. The topic, CO2 supplementation and yield and nutrient concentrations, is not novel, and the results are to have been predicted given earlier publications [e.g. reference 18], only the iron issue is different to other publications, but the authors do not give any reason for this effect.
I recommend returning to the authors, with a strong statement to indicate that the issues raised by myself be addresses. I attach a scanned annotated version of the manuscript.
In particular,
Line 28/29 this is incorrect. Maybe correct if you calculate content [content = weight x concentration per plant or similar], but you did not attempt to do this!!
Line 22 indicate what SPAD and atLeaf values represent.
Line 2/26 and other instances in text, YOU measure concentration not content. Be specific and write concentration.
Line 441 the improved water use efficiency is not only due to recycling of water, so indicate so.
Line 93 shape of holes? And spaced so far apart, this is most unusual practice.
Line 172 Table 1 suggests to the contrary of what is written.
Page 6 re-align the species titles.
Line 192 what is ‘greater size’? Very vague
Line 188 Section 3.3 why no effect on chard dry weight, this must be mentioned and discussed.
Line 211 surely it increases source size in leafy vegetables too??
Line 213/4 not for Swiss chard.
Line 217 reference [34] refers to ‘grown at either ambient or twice ambient concentrations of CO2’ so how are these values obtained for ppm CO2?
Lines 234-237 are not relevant to this study.
Line 249/250 any other species with similar effect?
Lines 285-305 information missing…
Line 310 name of publisher?
Line 338 strange to have a volume number in a symposium? Verify.
Line 350 and 352 one-page publications?
Line 361 abbreviate journal title.
Line 386 italics for journal title.

Reviewer 2 Report
Effect of Greenhouse CO2 Supplementation on Yield and Mineral Element Concentrations of Leafy Greens Grown Using Nutrient Film Technique
I write you in regards to manuscript # agronomy-717865 entitled "Effect of Greenhouse CO2 Supplementation on Yield and Mineral Element Concentrations of Leafy Greens Grown Using Nutrient Film Technique" which you submitted to the agronomy.
Authors need to follow the following instructions to improve this manuscript
The author should check page numbers before submission. Page 3, Line 109-: Experimental Setup (Author should provide the humidity of greenhouses) Page 4, Line 0-: Describe the measuring method of leaf mineral element. Page 1, Line 200-: Author should rewrite carefully. In this part, some portion wrote as like as Introduction. Please improve line 201-210, line 227-237, line 244-249, line 258-264, line Page 2, Line 285-291: What do you mean XX, YY, ZZ? Data should add: photosynthesis, stomatal conductance, calculated yield, chlorophyll (a, b, total) Data should recheck: N and Fe. References: should follow the journal guideline Please check carefully before resubmission.
I recommend to improve the manuscript and resubmit.

Round 2
Reviewer 2 Report
Accept after confirming the clean file
Author Response
Minor edits required
Line 66: basil is Ocicum basilicum
Response- Changed to Ocimum basilicum L.
Line 121: The value reported is relative humidity, not humidity
Response- Added word relative
Lines 150-151: p-values for 0.001 and 0.0001 are not reported in Table 1. Only need to report that p-value at 0.05 was reported.
Over use of the word "significant" mean values are either different or similar. Use of the word significant implies an order of magnitude, not a statistical inference. When used to discuss statistical differences, using the phrase significantly different is grammatically redundant.
Response- Removed the p values for 0.001 and 0.0001 in text. Also removed word significantly to avoid redundancy.
Line 205: Were HAF fans present in the greenhouse? Tip burn can be reduced with increased air movement.
Response- No the HAF fans were not present in greenhouse. Added line about its potential in discussion in line 267-269.
Line 295: Please indicate what the Fe level was in the nutrient feed.
Response- Indicated Fe level in nutrient feed.
Line 297-298: The use of HAF fans and air movement is an additional study parameter for future work
Response- Added recommendation for future work